# Influence of Spatial Accessibility and Environmental Quality on Youths' Visit to Green Open Spaces (GOS) in Akure, Nigeria

Obinna Justice Ubani [1,*] , Micheal Oloyede Alabi [2], Emmanuel Ndukwe Chiemelu [3], Andrew Okosun [1] and Chinwe Sam-Amobi [4]

[1] Department of Urban and Regional Planning, Enugu Campus, University of Nigeria, Enugu 410001, Nigeria; andy.okosun@unn.edu.ng

[2] Department of Urban and Regional Planning, Federal University of Technology, Akure 340110, Nigeria; moalabi@futa.edu.ng

[3] Department of Geoinformatics and Surveying, Enugu Campus, University of Nigeria, Enugu 400102, Nigeria; emmanuel.chiemelu@unn.edu.eg

[4] Department of Architecture, Enugu Campus, University of Nigeria, Enugu 400102, Nigeria; chinwe.sam_amobi@unn.edu.ng

[*] Correspondence: obinna.ubani@unn.edu.ng

**Abstract:** Although a regular visit to green open spaces has been recommended in curbing some sedentary lifestyle-associated health challenges, not much has been performed to uncover the factors that promote visits to such spaces in urban areas in sub-Saharan Africa. This research investigated the spatial accessibility and environmental quality factors that influence visits to green open spaces (GOS) by youths in the core area of Akure, Southwest Nigeria. The data were derived from a survey of 400 respondents aged between 18 and 35 years, ArcGIS software, v. Pro 20, and geographic information system (GIS) and analyzed using descriptive statistics and binary logistic regression analysis. The results revealed that the spatial accessibility predictors of visits to GOS among the youths were their age, street integration, social integration, and proximity of GOS to their homes, while the environmental quality predictors were the presence of a gymnasium in GOS and public facilities around such spaces. These findings are instructive in noting that to ensure social justice in access to GOS and the associated social and ecosystem services they offer, the planning, design, and management of green spaces should take cognizance of these predictors in meeting the needs of the youths who constitute a larger percentage of the urban population and thus encouraging them to regularly visit GOS in the study area and beyond.

**Keywords:** core area; green open space; environmental quality; spatial accessibility; social justice; urban areas

## 1. Introduction

The dynamics of urban living are influenced by global challenges, economic globalization, and improved computer technology. City residents have now imbibed sedentary lifestyles with reduced physical activities. A plethora of recent research revealed that individualistic youths were more vulnerable to sedentary behavior [1,2]. A sedentary lifestyle has generally been attributed to an increase in public health issues, being overweight, and ailments related to loneliness [3,4]. However, in this era of environmental development, resilient cities, park cities, spongy cities, and green cities have been identified as capable of providing solutions to the rising urban issues by contributing to aiming to building a healthier, safer, and more livable urban space and providing biocomfort in urban residential areas [5]. Following this, developing a viable urban green open space (GOS) has been proposed in this study.

A city's main public open space has specific varieties of value systems and manifold roles in promoting urban sustainable development [6]. The green open space visitation has

been recommended as a panacea to alleviate some sedentary lifestyle-associated public health challenges by providing diversified leisure resources, reducing exposure to environmental hazards, and encouraging physical activities [7,8]. The study has shown that due to the complicated nature of the world, people now seek solace in areas of nature [9]. Reference [10] explained that exposure to green spaces mitigates the risk of prolonged health conditions, such as respiratory diseases and obesity. Concurrently, green open spaces are progressively viewed as sites for 'curative' contact with nature since various environmental damages have posed serious threats to life and numerous habitats [11,12] and social rendezvous [13]. However, a common problem described in the body of literature is the inadequacy of green spaces that meet the needs of the different age cohorts of society [3,4]. References [14,15] pointed out that youths living in the city core suffer from the inadequacies of open space attributed to weakening physical conditions and mobility. Indeed, perceived lack of exposure to green space has been independently linked with overweight in the youths [2] and mental and physical health issues [16]. Following this, the United Nations General Assembly (2016) noted that the provision and usage of green spaces are integral parts of the Habitat III Agenda aimed at promoting human health and well-being in urban areas in the 21st Century and contributing to the attainment of the sustainable development goals [17] This provision of green open spaces will reduce the effect of global climate change which has been projected to cause changes in climate parameters and directly affect organism on earth [18].

Accessibility to the GOS is the right of every urban resident irrespective of class, age, gender, and age. However, it has been explained that the unique necessity of the youths to use green open spaces is not adequately integrated into the overall planning framework of towns [19]. Studies have shown that certain individual needs are often overlooked in planning green spaces; these form factors of users' needs [2,20]. This is very common in cities in sub-Saharan Africa [21,22]. As a result, a majority of urban residents have limited access to green open spaces, which affects the environmental quality of urban green open spaces [15]. Visitation of green open space essentially depends on its level of space accessibility [22] and environmental quality [23]. These could act as pull or push factors depending on the needs they meet. Essentially, accessibility to green open space and a variety of quality amenities, with quality of the environment, had been used and established as indicators of measurement in the planning design of GOS [24,25]. Characteristically, the spatial accessibility level denotes physical distance and obstacles to visiting open spaces [1]. Even though studies have established that the benefits of urban green spaces are appropriated by human beings if they are physically or visually accessible [26,27], the level of spatial accessibility has not been properly measured and integrated into the planning and design of urban green spaces, especially in urban areas in sub-Saharan Africa [22].

The review of extant literature revealed that several factors influenced the decision of the urban population to visit urban green open spaces. These include the quality of vegetation [23], maintenance and management [28]; closeness of open green spaces to the residents [29–31], street connectivity, adequacy, and safety issues), [32] availability of basic facilities [33,34] and demographic factors [15,35]. In Nigeria, previous studies have shown the factors that influence urban population visits to urban open green spaces include socioeconomic factors [26], perceived health [26], and social benefits associated with this [22].

Relating to the influence of the closeness of open green spaces to the residents and street connectivity on visits to open green spaces, there is a growing body of literature associating time travel and distance with streetscape design features and street networks [36]. One of the features of street networks is street integration, which has been described as an easy way to walk unimpeded through a street in a neighborhood [37]. Related to this is street connectivity, which describes how connected streets are [38] and the availability and directness of alternative routes from one location to another within a street network [39], which has been identified as a key contributory factor to spatial accessibility [39]. Given that street integration and connectivity are partly determined by the density and quality of

roads, the availability and quality of access roads leading to GOS have been identified as a determinant of visits to GOS [40].

In spite of the insight gained from the existing studies, there are limited studies on the youth visit to the GOS of the city core, especially in cities in the global South. Consequently, there is a gap in knowledge of the specific aspects of spatial accessibility and environmental quality with the most significant influence on the decision to visit open spaces among the youth in urban Nigeria. It is against this background that the current research aimed at determining the spatial accessibility and environmental quality factors influencing visits to green open spaces by youths in the core area of Akure, Southwest Nigeria. In the context of this research, youth refers to an individual of age between 18 years and 35 years. This is in line with the Nigerian National Youth Policy. The study was guided by two specific objectives, which were to:

1.  Determine the accessibility factors that can predict youths' visits to green open spaces in the study area.
2.  Determine the quality factors that can predict youths' visits to green open spaces in the core area of Akure, southwest Nigeria.

Given that everyone has the right to enjoy green open spaces and the social and ecological services associated with these, the study makes a contribution to knowledge in revealing the spatial accessibility and environmental quality factors that influence the visit of young people to GOS in the study area and suggesting how the planning and design of GOS will improve visitation and social equity in urban areas in the global South. In essence, this research is valuable in uncovering the strategies for enhancing the contribution of green open spaces to the social sustainability of urban environments.

## 2. Context of Study

Akure is a city situated in the South-Western part of Nigeria (Figure 1). It is situated between latitude $70°15'$ North of the Equator and longitude $50°15'$ East of the Greenwich Meridian (Figure 1b).

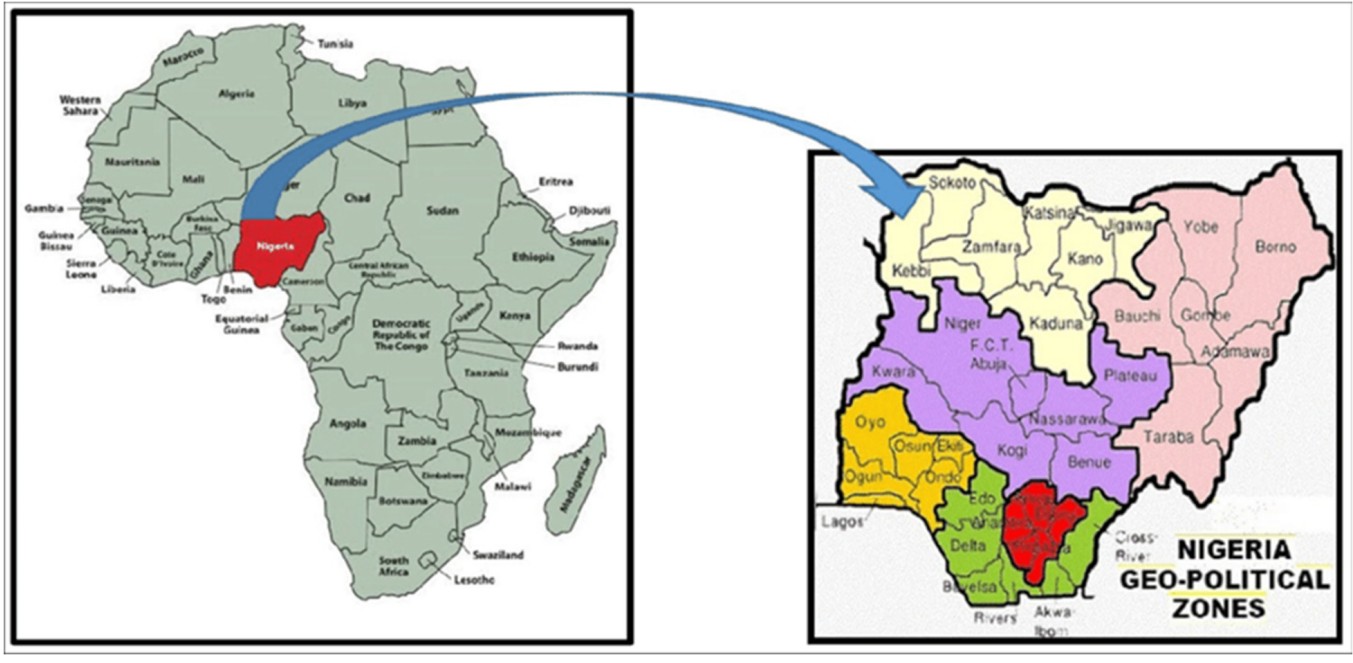

**(a)**

**Figure 1.** *Cont.*

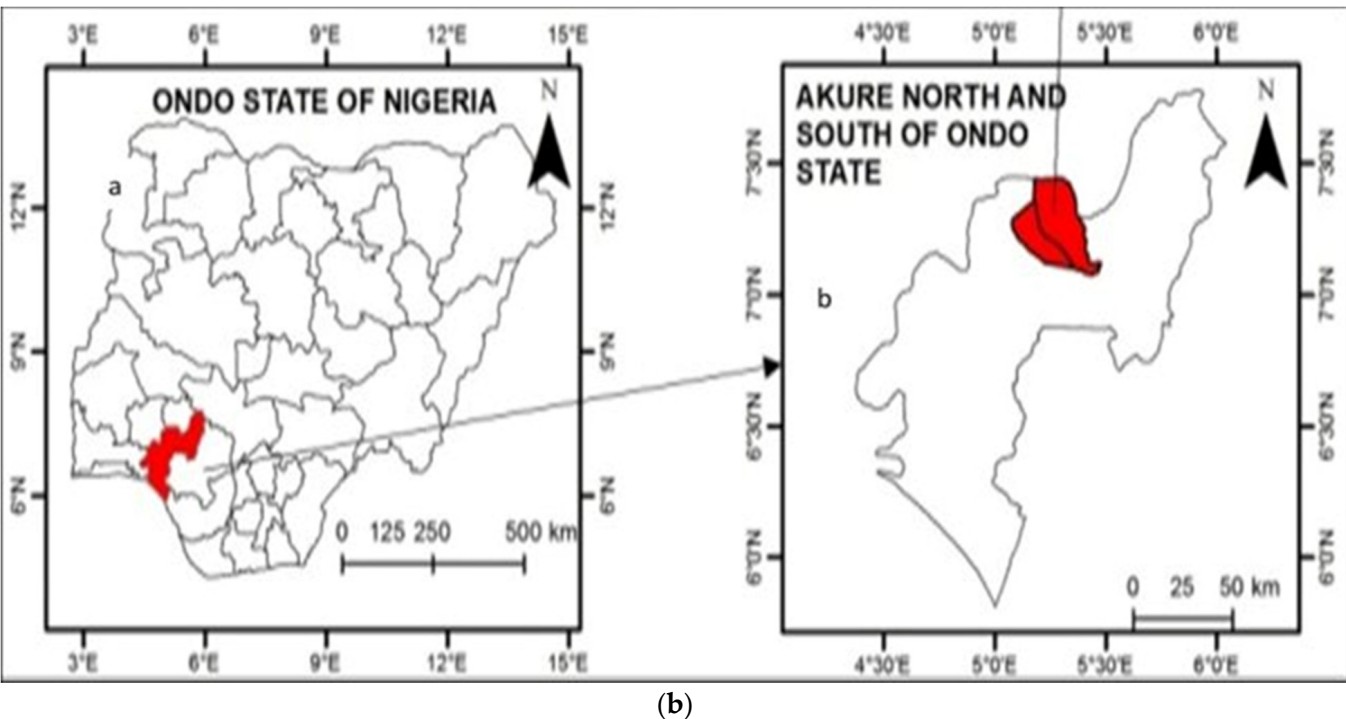

(**b**)

**Figure 1.** (**a**) Map of Africa showing Nigeria. (**b**) Akure in the context of Ondo State, Nigeria.

Akure is described as a medium-sized urban center that became the regional headquarters of Ondo Province in 1939. It was made the Ondo State capital in 1976 and, subsequently, a Local Government Headquarters. The city's population increased from 38,852 in 1952 to 71,106 in 1963. The 1991 National Population Census figure indicates that the city had a population of 239,124 persons [41]. This increased to 484,789 in 2006 and is estimated to be 756,386 in 2021, with a density of 490/sq km (1300/sq m).

There are many green open spaces used for recreation in the study area, which could be unorganized and organized. The unorganized green open spaces are mostly incidental and only function during festive periods (like Sallah, Easter, and Christmas); they are usually arranged by private individuals or organizations, temporarily with makeshift facilities provided. They do not function daily and throughout the year. Facilities provided are removed after the festive periods. This can be found in green fields around the Alagbaka area (e.g., the state Secretariat Park located in the Alagbaka area) (Figure 2). The organized green open space includes the *Ilula Recreation* ground located in the Sijuade area of Akure and the *Wottss Garden* in the Alagbaka area; these are privately owned and managed by individuals. The Oyemekun Rocks and Unity Village are also privately owned and located in the Champion district of the city (Figure 3). Others are the Ondo State Senior Staff Club (Figure 4) and the Ondo State Ministry of Agriculture Biological Garden (Figure 5), which are owned by the Ondo State Government and leased to a private individual for management. There is also the Pa Fasoranti Botanical Garden located inside the Federal University of Technology Akure Campus, which is owned and managed by the university authority.

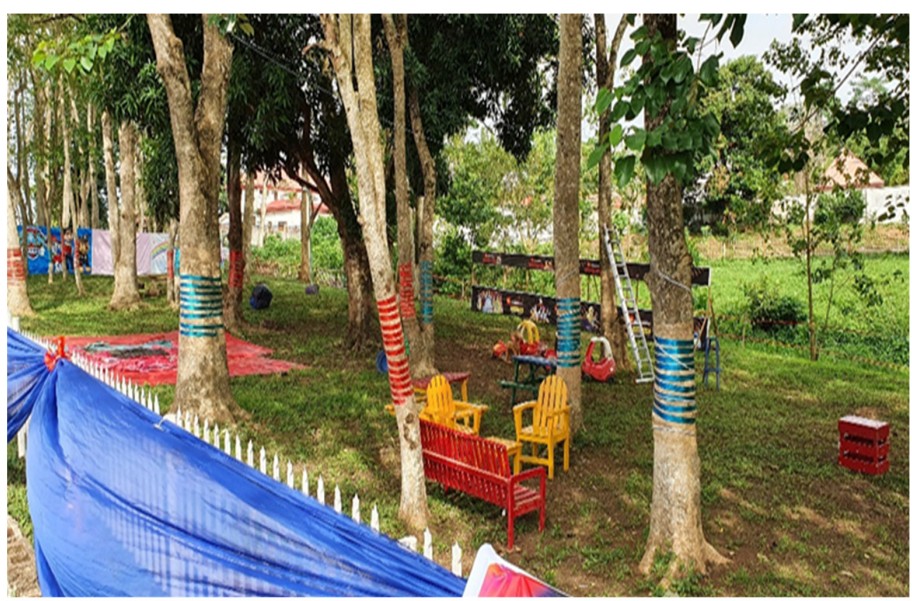

**Figure 2.** Green Open Space at State Secretariat Park in Alagbaka Area.

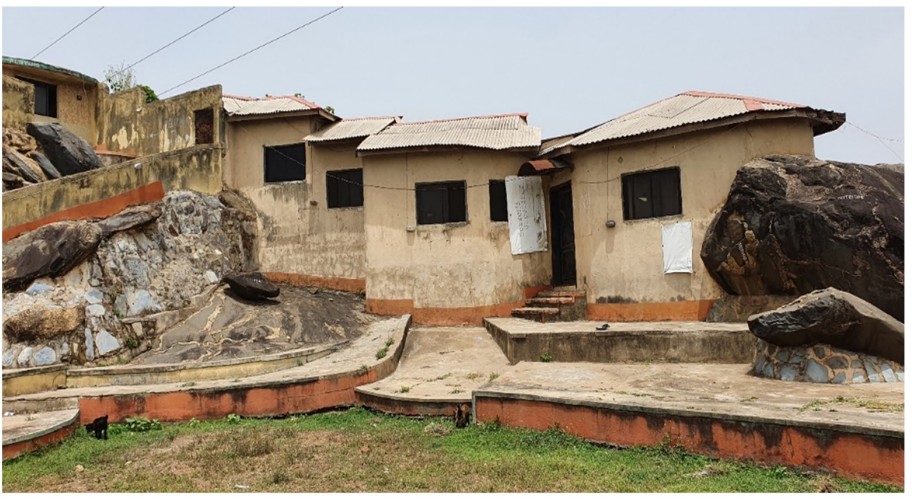

**Figure 3.** Chalets at Oyemekun Rocks and Unity Village.

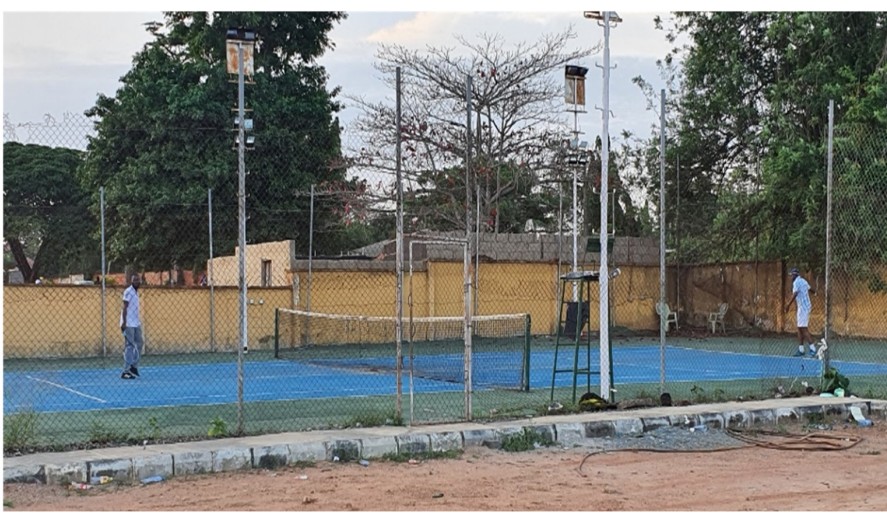

**Figure 4.** Lawn Tennis Court at Ondo State Senior Staff Club.

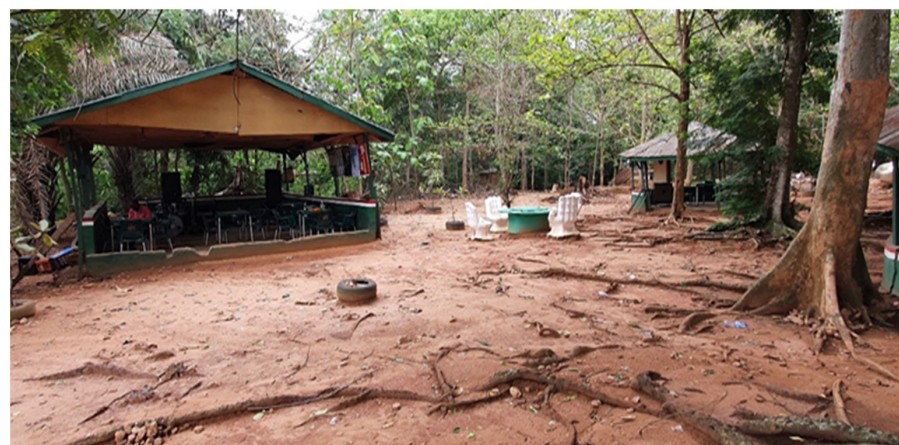

**Figure 5.** Chalets at Ondo State Ministry of Agriculture Biological Garden.

## 3. Review of Literature

### 3.1. Access to GOS as a Form of Social Justice

Accessibility to green open space is a vital attribute of the idea of social equity and rights [42]. This will depend on the individual's personal factors, community factors, natural factors, and naturalness [43]. The design of the green space (parks) must consider the rights and needs of the youths and other age cohorts to enhance their usage and appropriation of the services they offer. The right to the urban area is the common view that alludes to equal access to city resources and green spaces [15,44,45].

Social justice is a concept related primarily to justice and equality. A review of green space shows explicitly the idea of "fairness in access to GOS services" [15,46]. Social exclusion and injustice are now considered anathema from the perception of city living, linking to people who live in deprived neighborhoods [47]. Since the right to the GOS seems a corporate right and not personal, various studies have considered gender, diversity of race, age group, and economic class, and striving for a just solution to exposure to green spaces. However, unequal distribution of green open space can lead to marginalization from disadvantaged communities and certain age groups [35]. Therefore, the design and planning criteria of green open space ought to be improved to engender social equity for the youth.

Even though various studies have looked at the connection between the concept of social justice and the use of green open space by youths, most are related to the spatial distribution perspective [48]. In their study of Odense, Denmark, [49] found a link between gender and age, noting that men were far more likely to visit green spaces at least a few times a week. While [50] showed that proximity to parks or parkland influences the activity level of the younger (18–34) and older (55+ years) groups than in other age groups. Additionally, age is influenced by different constraints like facilities (quality and quantity), which reduces visits to urban green spaces [51]. However, [52] posited that the age of citizens was a strong determinant of visits to parks, with young people having more tendency to likely visit than older ones. Less emphasis was placed on other personal factors such as the needs of the youths in terms of quality of amenities, facilities for physical activities; the quality of green open spaces, like in areas of physical activities [53]. So also, accessibility and the standard of the open spaces are veritable parameters affecting the youths' ability to the right to green open spaces. Hence, social justice is highly influenced by factors of ease of reaching amenities and the type, quantity and quality of services available in the green open space. Inadequate access to green open spaces and services can make the youths less active than normal [54]. This constitutes and aspect of social injustice, which is contrary to the spirit of sustainable development goals (SDGs).

### 3.2. Factors Influencing GOS Visits

Earlier studies of open space have stressed the availability and distribution of open space [55], while [14] considered the importance of equality in the value, quality, and accessibility of open spaces. In a recent review, [15] highlighted the peculiarities of youth concerns about infrastructure and facilities, landscape features, accessibility, and maintenance when visiting open spaces. Figure 1 depicts the conceptual and theoretical framework for understanding the effects of factors of visits by youths to GOS. Based on the concept adopted by previous authors [15,53], the influential variables were separated into two main categories, and these include space accessibility factors (social integration, street connectivity, street integration, and proximity to home) and environmental quality factors (facility types, size of GOS, facility quality, public facility around GOS, and naturalness).

Furthermore, several studies have investigated the factors influencing visits to GOS. Among the two main dimensions of factors reported to influence visits to green open spaces is accessibility to their locations [11,22,27]. This includes factors such as the closeness of open green spaces to the residents' homes [15,29] and street connectivity [32]. The physical environmental quality of open green spaces [23]. These could act as pull or push factors depending on the needs they meet. In this context, accessibility refers to physical distance and obstacles to visiting open spaces, while environmental quality includes several factors, including quality of vegetation [23], maintenance and management [28], availability of basic facilities [34], and adequacy of the facilities, and safety and security of green open spaces [32]. Another group of factors identified as having an influence on visits to open green spaces is related to visitors' socio-demographic variables [15,26,35,56]. Table 1 is a summary of the accessibility and environmental quality factors linked to visits to GOS in urban areas as gleaned from the published literature.

**Table 1.** Accessibility and environmental quality factors influencing visits to GOS.

|  | Factors | Description | References |
|---|---|---|---|
| Accessibility | Social integration | Invitation from-<br>1. friends; 2. family; 3. peer | [49,53] |
|  | Proximity to home | >1 km from home<br>1 to 5 km<br>5.5 to 10 km<br><10 km | [36,37,57] |
|  | Street Integration | Space syntax theory variable, which shows the closeness of each segment is to all others under each definition of distance | [37,58,59]. |
|  | Connectivity of road leading to Open space road |  | [37,58,59] |
| Environmental Quality | Facility type | Playground, Sports field, Swimming pool, Basketball court, Tennis court, Race Tract, Restrooms, Drinking fountains, Benches, Picnic tables, Picnic shelters, Grills, Trash cans | [50,59]. |
|  | Size of GOS | open space size, larger open space, cold attracts more visits | [36,60] |
|  | Facility quality | Quality of facilities in open space. Good quality facilities may attract more visits to open space | [43,61]. |
|  | Public facilities around GOS | Convenient store, Shopping mall, City event hall and Banks | [62,63] |
|  | Naturalness | Clean air<br>Low noise<br>Vegetation coverage<br>Rich vegetation | [15,54] |

Source: Compiled by the Authors.

Following the evidence in the empirical literature reviewed here, a conceptual framework was developed for the study (Figure 6). The framework was developed from two main strands of literature. First is the literature on the spatial accessibility and environmental factors that influence youths' visits to open green spaces, while the second is the literature on visits to open green spaces as an aspect of social equity in urban environments. The basic assumption in the conceptual framework is that youths' visit to green open spaces in urban areas is a function of spatial accessibility measured using street integration, street connectivity, social integration, and proximity to home; the environmental quality of green open spaces measured using facility types and quality, size of GOS, the naturalness of GOS and public facilities around GOS. It is also assumed that any of these factors found to have a positive influence on youths' visits to GOS contribute to social equity, especially as it relates to access to social and ecological services associated with green open spaces in urban areas.

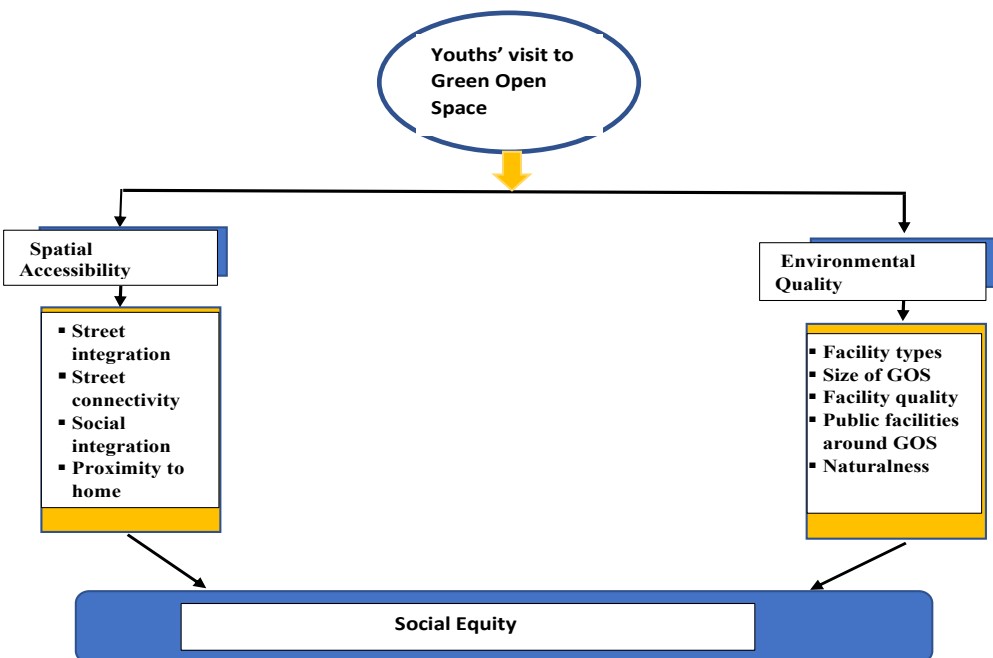

**Figure 6.** Conceptual framework of the study.

## 4. Materials and Methods

The research design adopted in this research is a combination of a cross-sectional survey and an experiment. This choice was based on the nature of the subject matter investigated and the research objectives. The human research population included youths aged (18–35) years, while the non-human research population comprised green open spaces within the core area of Akure town, Ondo State, Southwest region of Nigeria. The green open spaces that fall within the core area of Akure are the Oyemekun Rocks Recreation Center, FUTA Botanical Garden, Ilula Recreation Center, Ondo State Botanical Garden, and Senior Staff Club. The study sites were selected based on the nature of green open space (only organized open spaces were selected), then the ownership and management status. The organized green open spaces selected for the study were those privately owned and managed by individuals, those owned by the State Government and leased to a private individual for management, and those owned and managed by a university authority.

The minimum sample size was derived from the total number of visitors of 1500 per year, given from the combined register of the parks as revealed during a reconnaissance

survey. Based on this, Yamane's (1967) formula for the finite population was used to determine the minimum sample size for the survey. The formula is given as:

$$n = \frac{N}{1 + N(e)^2} \tag{1}$$

where n = Sample size; N = Total number of visitors; e = level of precision = 5% (0.05), at 95% confidence level and precision of 5% = 0.05 significance level

$$n = \frac{1500}{1 + 1500(0.05)^2} = 315.78 \approx 316 \tag{2}$$

A total of 83, representing 28% of the calculated sample size, were added for non-responses or lost questionnaires. This translated to a minimum sample size of 399 participants. This means that the lowest number of participants in the survey in Oyemekun Rocks Recreation Center, FUTA Botanical Garden, Ilula Recreation Center, Ondo State Botanical Garden, and Senior Staff Club should be 399 persons.

The principal data-gathering instrument for the survey was a pre-tested questionnaire designed by the researchers. It was designed to collect data on social integration, facility quality, facility type, and naturalness. Table 2 shows some of the variables and measurement scales included in the questionnaire.

**Table 2.** Variables and Measurement scales.

| Variable | Factors | Evaluation Elements | Measurement | Assessment Scale |
|---|---|---|---|---|
| Dependent variables | Visitation | | | Yes = 2 No = 1 |
| Independent variables | Accessibility | Social integration | Friend invite/ family Invite/ peer invite/ | Dummy variable (1 = yes, 0 = none) |
| | | Proximity to home | >100 m 100–250 m 250 m–<500 m | 1= 10–25 min 2 = 26–50 min 3 = 51–<61 m |
| | | Street Integration | Calculated using DepthmapX software, Version 0.6.0 | Ratio |
| | | Street Connectivity | Calculated using DepthmapX software | Ratio |
| | Environmental Quality | Facility type | Playground, Sports field, Baseball field, Swimming pool, Basketball court, Tennis court, Race Track, Drinking fountains, Picnic tables, Picnic shelters, Grills, Trash cans | Nominal (1 = yes, 0 = none) |
| | | Size of GOS | Calculate using ArcGIS. | Ratio (M$^2$) |
| | | Facility quality | In situ observation, quality of facilities in open space. Determines visits to open space | Nominal1 = yes, 0 = none) |
| | | Public facilities around GOS | Using ArcGIS | Ratio: (Number of community facilities within the service area) |
| | | Naturalness Clean air Low noise Vegetation coverage Rich vegetation Good natural scenery | In situ observations | Ordinal Scale: None/poor/fair/good (0, 1, 2, 3) None/poor/fair/good (0, 1, 2, 3) None/poor/fair/good (0, 1, 2, 3) None/poor/fair/good (0, 1, 2, 3) None/poor/fair/good (0, 1, 2, 3) |

The other data gathering instruments used were observation schedule, ArcGIS software, and geographic information system (GIS). These were used in spatial analysis and environmental auditing.

The research was carried out between January 2021 to July 2022. Each of the study sites was checked bi-monthly within the study time to check its naturalness. Prior to this, a pilot survey was performed to ensure that the respondents had comparable standards of assessment considering the standard of open space character investigated. In the questionnaire survey, the respondents were randomly selected from each green space area: Oyemekun Rocks Recreation Center, FUTA Botanical Garden, Ilula Recreation Center, Ondo State Botanical Garden, and Senior Staff Club. A purposive sampling technique was employed in determining the participants, and only the users of the spaces within the age bracket of 18 years and 35 years were given a copy of the questionnaire to fill out. The services of four Research Assistants were engaged to count the number of youths in each of the open spaces every 20 min between 10:00 a.m. and 1:00 p.m. on weekdays and 6:30 a.m. to 1:00 p.m. on weekends. This was performed for two weeks on a bi-monthly basis. In line with the methods elucidated by [15], the aggregate population of youths was used for the recreation area visits. The number of youths in each session counted was used for further inquiries about their visits to the open spaces with the use of a questionnaire (Table 2). A total of 400 copies of the questionnaire were distributed and retrieved. Since the respondents were found on the spot and the questionnaire was administered and collected immediately, the response rate was 100%.

In collecting data on the spatial accessibility factors, spatial analysis was used to elucidate data on road accessibility. The street integration and connectivity were derived from axial maps after space syntax analysis. The proximity from home was extracted from the options selected on the questionnaire posited to the participants. Specifically, the space syntax and geographic information system (GIS) technique were also employed through data collected from the digitized open road map of Akure. In order to calculate the GOS integration variable, the space syntax method was used; this was apt in measuring open space residents' accessibility, as explained in previous studies [37,64]. Topological descriptors, which are defined as 'the least set of such straight lines which passes through each convex space and links all axials' [37,58], were calculated in order to quantify spatial structures based on the axial map. In doing this, all roads were assigned values based on the output from the space syntax. Again, the space syntax software, called DepthmapX, was also used to compute the various street centerlines [65]. Subsequently, open spaces were designated with separate integration scores based on streets, and their entrances were seen through an on-site survey (Figures 7 and 8). Open spaces with various entrances are allocated the greatest integration value. A high integration score depicts easy street access.

The accessibility factors, such as street integration and street connectivity, were obtained from spatial maps and network analysis carried out using DepthmapXsoftware 2015 [58]. The study area was buffered in the traditional circular form to analyze the open space, which seems very suitable since we had a uniformly flat terrain similar to what was obtained in the studies [66,67]. The service area for youths' activity space was set at a radius of 500 m buffer as recommended in previous research works (see for examples [68,69]. The proximity variable was selected from the alternative given on the questionnaire on the gap between open spaces and closet residents' homes. The information on the location of public facilities was elicited from the digitized map of Akure. The number of public facilities around open space and the total size of other open spaces located within the network buffer was calculated using a 500 m radius around GOS. For each open space, the variables (Table 2) were independently evaluated using the method in a previous study by [70].



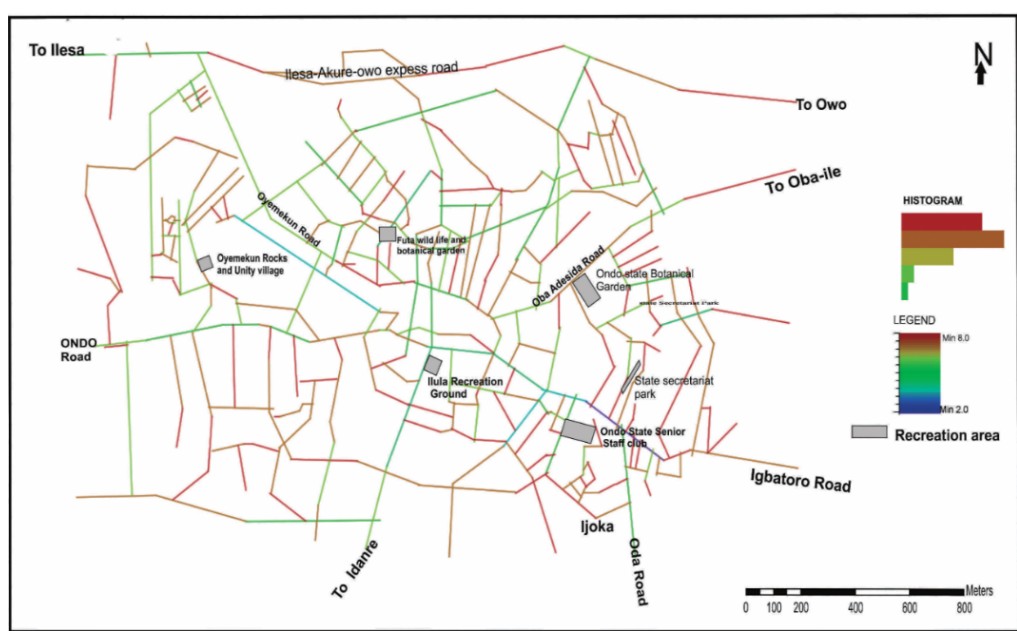

**Figure 7.** Axial line map showing road connectivity and GOS in the study area.

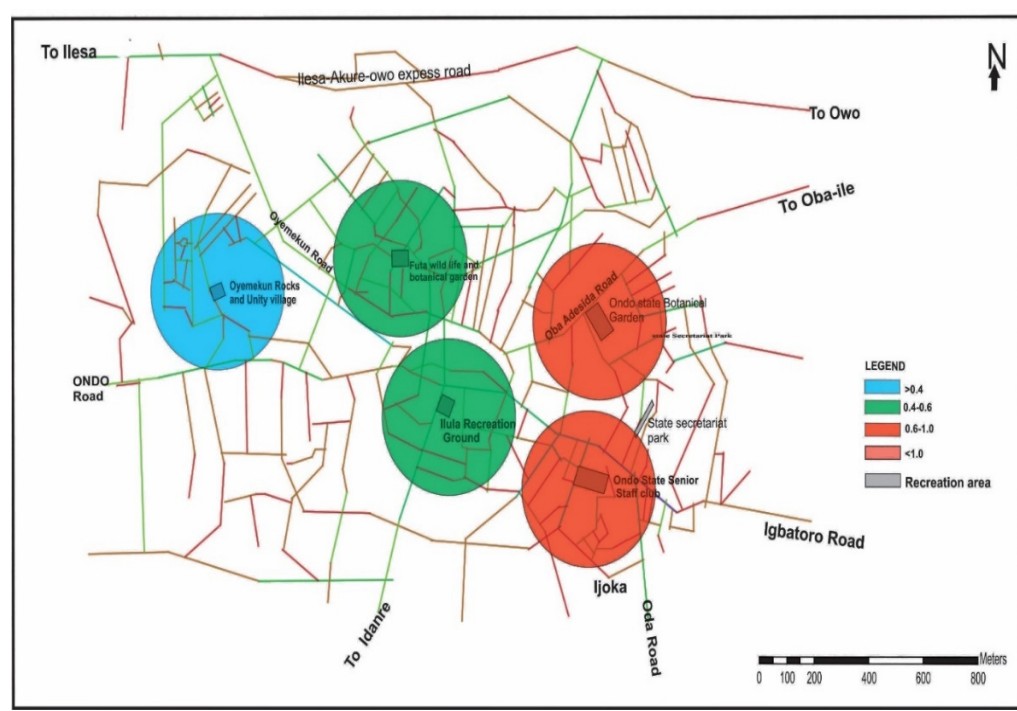

**Figure 8.** Axial line map showing the local Integration score of each recreation area in the buffer zones of the study.

The environmental quality factors investigated include the size of GOS, facility quality, public facilities around GOS, and naturalness. Four trained Research Assistants were engaged to elicit these variables and factors using the environmental audit of the surroundings GOS. ArcGIS software was used to calculate the size directly from the Akure digitized map.

The data were processed and analyzed using the SPSS software, v. 22. Two key types of analyses were conducted. The first was descriptive analysis using descriptive statistics, while the second was inferential analysis using the binary logistic regression model. In the first type of analysis, final scores for each variable were obtained by computing the mean

score of evaluation given by the various field research assistants. In the situation that the difference between the scores does not exceed 1 or if the difference is greater than 1, then the evaluation was repeated. The findings in this study depicted an average agreement between the two (kappa = 0.550, $p < 0.001$) and three (Fleiss kappa = 0.401, $p < 0.001$) research assistants' evaluations. To identify the factors influencing youths' visits, the binary logistic regression model was used. The dependent variable was the cumulative visits, while the independent variables were data for quality and accessibility factors (Table 2). The linear regression model, which sought to predict the influence of spatial accessibility factors on youths' visits to GOS, was expressed mathematically with the formula below:

$$Y = e^{\hat{}} (b_0 + b_1{}^*x)/(1 + e^{\hat{}}(b_0 + b_1{}^*x)) \tag{3}$$

where

Y—predicted output
x—input value
$b_0$—the bias or intercept term,
$b_1$—the coefficient for the single input value (x).

$$Y(visits) = 0.164X_1 + 0.524X_2 + 329X_3 - 0.602X_4 \tag{4}$$

where $X_1$ is the age group (youth), $X_2$ is street integration, $X_3$ is social integration and is $X_4$ proximity to home.

Similarly, the prediction regression model of environmental quality factors of the youths' visits to GOS is mathematically represented with this formula in Equation (1):

$$Y = e^{\hat{}}(b_0 + b_1{}^*x)/(1 + e^{\hat{}}(b_0 + b_1{}^*x)) \tag{5}$$

where Y is the predicted output is the input value, $b_0$ is the bias or intercept term, and $b_1$ is the coefficient for the single input value (x)

$$Y(visits) = 0.033X_1 + 0.188X_2 + 1.697X_3 + 0.205X_4 + 0.101 X_5 \tag{6}$$

where $X_1$ is the drinking fountain, $X_2$ is the size of GOS, $X_3$ is the gymnasium, $X_4$ facility quality, and $X_5$ public facilities around the GOS.

The results are presented using texts, tables, and graphical illustrations. The questionnaire used for the study was subjected to face and content validity; Cronbach alpha was also used to test its reliability as it scored 0.819, which shows that it was significantly adequate for the study.

## 5. Result

### 5.1. Spatial Accessibility Factors Associated with Youths' Visits to GOS

The groups of factors investigated are accessibility factors, the number of public facilities around GOS, and street connectivity. The accessibility factors consist of age group(youth), which was derived from the use of a questionnaire, and the result shows that based on the sampling technique adopted, participants in the survey were mainly those within the age bracket of 18 years and 35 years. As regards social integration, which was investigated using the reasons for visiting open spaces, the results indicate that most of them visited the spaces because they were either invited by their friends/peer groups or family members.

Regarding public facilities around GOS, findings of the environmental audit and analysis of the activity areas indicate a linear location pattern along the most locally integrated streets, which tend to be congested at the intersections of the most globally integrated roads where the local integration is low. This is indicative of a successful, vibrant shopping and business area, which coincides in this research with areas of public facilities. Hence, the location and number of public facilities around were found as five for Oyemekun

Rocks, seven for FUTA Botanical Garden another seven for Illula Recreation area, eight for the Ondo State Botanical Garden, and ten for the Senior Staff Club. This means that the largest number of public facilities are around the Senior Staff Club, followed by the Ondo State Botanical Garden and the least by Oyemekun Rocks, respectively.

Regarding street connectivity, Figure 7 presents the axial map of the study area, depicting the street connectivity of green open spaces (GOS).

Figure 8 shows areas and their local integration scores. High local spatial integration scores represent the areas of effective pedestrian-based activity places. It is often depicted using a dense structure on the street network within a short distance in the locale of hectic human activities. It shows that the ease of movement in the Oyemekun Unity Rocks falls within the category of low integration 0.4–0.6, which translates to areas of less successful pedestrian movement, while the FUTA Wildlife Botanical Garden and the Ilula Recreation Centers fall within the category of 6.0–4.0, depicting the area of moderate integration and moderate pedestrian ease of movement. The Ondo State Botanical Garden shows a higher integration of 0.6–1, and the Senior Staff Club indicated the highest integration of greater than 1, which depicts the area of most successful pedestrian movement.

Figure 8 also shows the segment street integration in the study area buffered around a radius of 500 m of the selected GOS. The little metric radius specifies the to-movement potentials for pedestrians, while the bigger metric radius specifies the to-movement potentials for larger vehicles. Areas with very high integration can be seen to have the highest to-movement potentials on a local scale and vice versa. The outcome of the analyses also shows the level of integration coincides with the vicinity of the public amenities. This sort of configuration is very attractive to pedestrians. Hence, a high population visits these areas.

### 5.2. Spatial Accessibility Predictive Factors

A binary logistic regression model was used to ascertain the association of age, street integration, street connectivity, social integration, and proximity to home with the likelihood of youths' visitation in the green open space (GOS). An initial analysis indicated that the assumption of multicollinearity was satisfied (tolerance = 0.97). The result of the model suggested statistically significant $\chi^2(5, N = 400) = 29.329$, $p = 0.000$, suggesting it could distinguish between visitation and non-visitation in GOS. The model explained between 7.1% (Cox and Snell R squared) and 11.1% (Nagelkerke R square) of the variance in the dependent variable and correctly classified 81.8% of cases. The Hosmer-Lemeshow was used to test the hypothesis that predictions made using the model fit perfectly with observed group memberships. The observed frequencies with those of expected frequencies were compared under the linear model with the aid of chi-square statistics. The chi-square result suggests that the data fit the model well was non-significant, $\chi^2(8) = 15.606$, $p < 0.005$. Hence, there is no significant difference observed between the predicted and the observed. (Hosmer and Lemeshow Test). The model explained 48.0% of the variance in visitation and correctly classified 79.8% of cases (Appendix A).

Results of the regression model revealed that age, street integration, social integration, and proximity of the home to the GOS within 500 m could attract more youths to visit green open spaces. Again, a rise of 1 m² of open space may increase 1.207 the likelihood of youths' visits. Furthermore, an increase in the number of gymnasiums in open spaces within 500 m will increase youths' visits by 5.459 times. In addition, an increase in public facilities of one unit around GOS may increase the likelihood of youths visiting 1.106 times. The age factor is well associated with the model at $p = 0.044$. The odds of visits to the GOS by the youth are 1.178 times more likely to occur with a 95% confidence interval (CI) of 0.775 to 1.790. Similarly, the factor of street integration is well associated with the model at $p = 0.000$. It depicts that youths 'visits to the GOS are 1.689 times more likely to occur due to the influence of street integration, with a 95% CI of 1.311 to 2.175. The factor of social integration also depicts a social integration with the model at $p = 0.028$, with odds of youths

visiting the GO 1.390 times more likely to occur due to the influence of social integration with a 95% CI of 1.035 to 1.867 (Table 3).

**Table 3.** Logistic regression of accessibility factors.

| | B | S.E | Wald | df | Sig | Exp (B) | 95% C.I (B) | |
|---|---|---|---|---|---|---|---|---|
| | | | | | | | Lower | Upper |
| Age group (Youth) | 0.164 | 0.214 | 0.586 | 1 | **0.044** | **1.178** | 0.775 | 1.790 |
| Street integration | 0.524 | 0.129 | 16.484 | 1 | **0.000** | **1.689** | 1.311 | 2.175 |
| Social integration | 0.329 | 0.150 | 4.798 | 1 | **0.028** | **1.390** | 1.035 | 1.867 |
| Street connectivity | −0.203 | 0.211 | 0.924 | 1 | 0.336 | 0.817 | 0.540 | 1.234 |
| Proximity to home | −0.602 | 0.242 | 6.211 | 1 | **0.013** | 0.548 | 0.341 | 0.879 |
| Constant | 0.716 | 0.892 | 0.645 | 1 | 0.422 | 2.047 | | |

Variable(s) entered on step 1: Age_group, street integration, soc_integration, Street connectivity, proximity home.

However, the factor of proximity to the home shows a significant association with the model at $p = 0.013$, but the odd ratio is lower than 1 (i.e., 0.341 to 0.879), hence suggesting that an increase of 1 m$^2$ of distance from home around open space within 500 m may reduce 0.548 times the likelihood of visit in the GOS. However, street connectivity does not show a good association with the model at $p = 0.336$. It shows less probability of change in the occurrence of visits in GOS (Table 3).

### 5.3. Environmental Quality Predictive Factors

The environmental quality factors investigated include facility types (swimming pool, lawn tennis, football field, picnic tables, drinking fountains, grills, trash can, picnic shelter, race track, gymnasium), size of GOS, facility quality, public facilities around GOS (banks, shopping malls, event centers, markets), naturalness (clean air, low noise, rich vegetation). Data presented in Table 4 shows clear statistics of green space quality and visits. The table shows the poor quality of the football field, race track, and gymnasium. In contrast, there are good quality elements found as picnic tables, size of GOS, clean air, and rich vegetation.

**Table 4.** Descriptive Statistics of Environmental Quality Factors of GOS.

| Factors | Categories | Mean | Standard Dev. | Min | Max |
|---|---|---|---|---|---|
| **Facility type** | Swimming pool | 0.1275 | 0.33395 | 0.00 | 1.0 |
| | Lawn tennis | 0.3750 | 0.48473 | 0.00 | 1.0 |
| | Football field | 0.0000 | 0.0000 | 0.00 | 0.0 |
| | Picnic tables | 1.0000 | 0.0000 | 1.00 | 1.0 |
| | Drinking fountains | 0.6250 | 0.48473 | 0.00 | 1.0 |
| | grills | 1.0000 | 0.0000 | 1.0 | 1.0 |
| | Trash cans | 1.0000 | 0.0000 | 1.0 | 1.0 |
| | Picnic shelter | 0.8750 | 0.33113 | 0.0 | 1.0 |
| | Race track | 1.0000 | 0.0000 | 1.0 | 1.0 |
| | Gymnasium | 0.0750 | 0.26372 | 0.0 | 1.0 |
| **Size of GOS** | | 2.5000 | 0.69684 | 1.0 | 3.0 |
| **Facility quality** | | 1.7825 | 0.55765 | 1.0 | 3.0 |
| **Public facilities around GOS** | | 2.8450 | 0.79218 | 2.0 | 4.0 |
| **Naturalness** | Clean air | 1.0150 | 0.12170 | 1.0 | 2.0 |
| | Low noise | 1.1450 | 0.35254 | 1.0 | 2.0 |
| | Rich vegetation | 0.8750 | 0.33113 | 0.0 | 1.0 |

Logistic regression was used to establish the association of quality factors of swimming pool, lawn tennis, drinking fountain, picnic, size of space, gymnasium, facility quality, clean air, and low noise. However, the factor of rich vegetation was eliminated due to redundancy. An initial analysis indicated that the assumption of multicollinearity was satisfied (tolerance = 0.95). The result of the model suggests that it was statistically significant $\chi^2$ (10, N = 400) = 62.998, $p$ = 0.000, suggesting it could distinguish between visitation and non-visitation in GOS. The model explained between 14.6% (Cox and Snell R squared) and 23.0% (Nagelkerke R square) of the variance in the dependent variable and correctly classified 79.8% of cases. The Hosmer-Lemeshow tests the null hypothesis that predictions made using the model fit perfectly with observed group memberships. The observed frequencies with those of expected frequencies were compared under the linear model with the aid of chi-square statistics. A non-significant chi-square indicates that the data fit the model well, $\chi^2(6)$ = 2.387, $p$ < 0.000. Hence, there is no difference between the observed and the predicted model (Hosmer and Lemeshow Test). The model explained 88.1% of the variance in visitation and correctly classified 79.5% of cases.

Results of the regression analysis (Table 5) indicate that providing a gymnasium and public facilities around the GOS within 500 m could attract more youths to visit green open spaces.

**Table 5.** Logistic Regression of Quality Factors.

| | B | S.E | Wald | df | Sig | Exp (B) | 95% C.I (B) | |
| --- | --- | --- | --- | --- | --- | --- | --- | --- |
| | | | | | | | Lower | Upper |
| Swimming pool | 18.897 | 40,188.779 | 0.000 | 1 | 1.000 | 16.108 | 0.000 | . |
| Lawn tennis | 20.013 | 4019.156 | 0.000 | 1 | 0.996 | 49.159 | 0.000 | . |
| Drinking fountain | 0.033 | 1.062 | 0.001 | 1 | 0.975 | 1.034 | 0.129 | 8.287 |
| Picnic shelter | 38.396 | 40,389.166 | 0.000 | 1 | 0.999 | $4.735 \times 10^{16}$ | 0.000 | . |
| Size | 0.188 | 0.714 | 0.070 | 1 | 0.792 | 1.207 | 0.298 | 4.891 |
| Gymnasium | 1.697 | 0.753 | 5.074 | 1 | **0.024** | **5.459** | 1.247 | 23.900 |
| Facility quality | 0.205 | 0.298 | 0.472 | 1 | 0.449 | 1.227 | 0.684 | 2.199 |
| Clean air | −0.803 | 0.987 | 0.662 | 1 | 0.416 | 0.448 | 0.065 | 3.099 |
| Low noise | −0.006 | 0.517 | 0.000 | 1 | 0.991 | 0.994 | 0.361 | 2.741 |
| Public facilities around GOS | 0.101 | 0.452 | 0.050 | 1 | **0.044** | **1.106** | 0.456 | 2.682 |
| Constant | −37.805 | 40,389.166 | 0.000 | 1 | 0.999 | 0.000 | | |

Variable (s) entered on step 1: swimming pool, lawn tennis, drinking fountains, picnic shelter, size, gymnasium, facility quality, clean air, low noise, Public_facilities_around.

An increase in gymnasium numbers around open spaces within 500 m may increase youths' visits 5.459 times. In addition, an increase in public facilities of one unit around GOS may increase the likelihood of youth visits 1.106 times. However, the value for the swimming pool shows a significance of 1 and that of Lawn tennis 0.996, at $p$ = 0.000, C. I, which indicates no activity, but statistically, it only indicates a lack of data since all respondents attested that there were no swimming pools in all the green spaces and only one recreation center had a lawn tennis court in all the GOS visited.

## 6. Discussion

This study aimed at determining the spatial accessibility and environmental quality factors influencing visits to green open spaces by youths in the core area of Akure, Southwest region of Nigeria. Some of the key findings of this research have been identified for further discussion. The first finding is that age, street integration, social integration, street, and proximity to home were the most significant accessibility factors influencing visits to GOS for the youths in the core area of Akure town, Southwest region of Nigeria. This result lends credence to the findings by [62], who reported that age was a strong influence on the visitation of GOS. This result was well expected because the youths are the most active in the city and have a very high tendency to engage in various physical activities, including visits to green open spaces to undertake various activities such as recreation, relaxation, social interactions, and the like. There is also the street integration factor, which is a strong variable as it determines the ease of movement. It is an established fact that young people

need ease of movement to carry out physical activities like cycling and walking, which are usually stifled in the dense city core due to a lack of provision of bicycle tracks or pedestrian walkways in road design in many urban areas in the global South. Specifically, the influence of street integration on youths' visits to recreational spaces was made clear in this research via the space syntax measurement, which revealed that a high and increased integration value is an indication of open spaces positioned in streets with an enclosed connection to other streets, and hence, increasing ease of movement. The implication of this is the frequent visits by youths to open spaces that are located on streets in areas with high integration values. Even though this finding is synced with that by [15,37], it is in contrast to normal expectations for connectivity and could be attributed to the high integration values of major roads, which are characterized by heavy vehicular traffic. There is also the challenge of crossing these road junctions and streets, which is risky for youths, mainly for those with low mobility [70].

Social integration was also identified as a strong factor in visits to GOS, as it sometimes determines what spurred the individual to visit the open green spaces. In support of the findings of previous studies [49,53], the current research also found that invitations by peers had the highest reason than invitations by family members for visits to GOS in the study area. Proximity to homes was also found to be strongly associated with visits to green open spaces among youths in the core area of Akure city. This result might have been due to the fact that the current research was focused on youths, unlike the other research that focused on older adults. It is also possible that the insecurity situation in the country and the recent outbreak of COVID-19 could be attributed to the youths not wanting to stay far away from their homes. This line of thinking resonates with the finding by [32] that safety and security issues have an influence on visiting green open spaces by urban residents. Even though this is consistent with the findings of previous studies [36,37,57], it seems to be contrary to the finding that distance to home was not significantly associated with visits to green open spaces [71]. Contrary to the evidence in the literature suggesting that street connectivity plays a positive role in visits to urban green spaces [32,39], street connectivity was found not to be significantly associated with youths' visits to GOS in the current research. This could be attributed to the high-density city core where the youths visit GOS within walking distance from their homes. This is contrary to what is expected in low-density urban areas, with good access, single-use developments, and widespread buildings, as explained by [69].

The second finding is that the two environmental quality factors associated significantly with youths' visits to green open spaces are the presence of a gymnasium and public facilities around the GOS. This finding on the gymnasium seems to provide support to previous studies indicating that facility type has an influence on visits to green open spaces (see [43,50,62]. Arguably, this can be linked firstly to the fact that consideration is usually given to different areas of urban green space in terms of leisure resources, facilities, and quality, as previously reported by [43] and secondly, because youths frequently use fitness facilities, hence the importance of the gymnasium in open spaces. In addition, the presence of public facilities around open spaces has the tendency to increase youth's prospect of visiting such spaces for relaxation after visiting close public facilities like banks, markets, and shopping malls, as previous researchers had reported (see [62,72]).

Although facility quality is another factor found to be significantly associated with visits to GOS [43,61], this did not emerge as a significant predictor of visits to green open space by youths in the present research. Similarly, contrary to the findings by [36,60] suggesting that the size of green open space was a determinant of visits to such spaces, this was not identified as a predictor of visits of youths to GOS in the study area. Notably, some of the variations in results between the current study and previous ones can be linked to the differences in the age groupings and cultural backgrounds of the participants. In any case, from the findings of this research, it is evident that among significant factors, street integration, social integration, the closeness of GOS to homes, and the presence of

gymnasiums and vital public facilities seem to be the most significant factors that drive youths' use of open spaces.

## 7. Conclusions and Study Implications

The study investigated the spatial accessibility and environmental quality factors influencing visits to green open spaces by youths in the core area of Akure, Southwest region of Nigeria. In line with the findings, the following conclusions are made. The first conclusion is that the age of the participants, street integration, social integration, and proximity of their homes to green open spaces were the most significant accessibility factors that influenced their visits to GOS in the core area of Akure town, Southwest region of Nigeria. The implication of this is that the ease of movement (street integration) is no doubt an important issue in promoting visits to green open spaces in the core area of cities. In view of road congestion and traffic issues within most cities in sub-Saharan Africa, urban planners should design pedestrian-friendly networks to open spaces and close them close to residents' homes in close proximity. The development of a proper urban design concept will make active green space possible [73]. Furthermore, urban planners and policymakers should consider locating green open spaces in a well-planned part of the city in proximity to the homes of the younger residents.

The second conclusion is that the two environmental quality factors significantly associated with youths' visits to green open spaces were the presence of a gymnasium and public facilities around the GOS. The implication of this is that planning authorities should partner with private companies to provide fitness facilities for physical activity to enhance their health benefits, as previously suggested by [32], even though youths may be more prone to engaging in interactive activities and social networks in a virtual environment on the internet, leading to a sedentary lifestyle with associated health implications, green open spaces with a gymnasium and public facilities around them can for engendering physical activities as noted by [74]. Therefore, open spaces that offer more innate space for group events and physical activities would be ideal and more attractive to youths.

The findings of this research are very informative for planners and managers of urban green open spaces in that they should take cognizance of the special needs of the youths in the design and planning of GOS to encourage their visits to such spaces. To achieve these accessibility and quality requirements, the GOS environment should be inclusive for the youths as opposed to other literature where older adults are seen as the vulnerable group whose needs should be prioritized to the detriment of others where health and socioeconomic status are considered [4,15]. Therefore, inclusiveness and equality are important considerations for youths regarding their use of the GOS [43,75].

In sum, it is expected that the findings of this study, as they relate to the predicted model of youths' visits to GOS, will assist urban planners and policymakers to understand and appreciate the impact of the various factors that influence youths' visits to open spaces and leverage on this in the future design and planning of green open spaces in a high-dense cities and urban environment in cities in the global South. Among other benefits, this will encourage youths to visit GOS, enhance the youths' right to access open space and promote social equity in enjoying the ecosystem services provided by green open spaces in urban built environments.

This study recognized that there are other variables that influence youths' visits to open spaces that were not captured in the work; therefore, this presents the essence for further studies; even situational circumstances are capable of influencing these visits. Hence, further research in that regard will be beneficial.

**Author Contributions:** Conceptualization, O.J.U. and M.O.A.; methodology, M.O.A.; software, M.O.A.; validation, M.O.A., C.S.-A. and E.N.C., A.O.; formal analysis, O.J.U., M.O.A.; investigation, A.O.; resources, O.J.U., C.S.-A. and E.N.C., A.O.; data curation, M.O.A.; writing—original draft preparation, M.O.A.; writing—review and editing, O.J.U. and M.O.A.; visualization, A.O., E.N.C.; supervision, O.J.U.; project administration, O.J.U.; funding acquisition, O.J.U., C.S.-A., E.N.C., A.O. All authors have read and agreed to the published version of the manuscript.

**Funding:** This research received no external funding.

**Institutional Review Board Statement:** Not applicable.

**Informed Consent Statement:** Informed consent was obtained from all subjects involved in the study.

**Data Availability Statement:** Not applicable.

**Conflicts of Interest:** The authors declare no conflict of interest.

## Appendix A. Regression for Accessibility and Environmental Quality Factors

**Table A1.** Case Processing Summary.

| Unweighted Cases | N | Percent |
|---|---|---|
| Selected CasesIncluded in Analysis | 400 | 98.3 |
| Missing Cases | 7 | 1.7 |
| Total | 407 | 100.0 |
| Unselected Cases | 0 | 0.0 |
| Total | 407 | 100.0 |

If weight is in effect, see the classification table for the total number of cases.

**Table A2.** Classification Table a,b.

| Observed | Predicted | | Percentage Correct |
|---|---|---|---|
| | **Visit** | | |
| | **No Visit** | **Visit** | |
| Step 0   visit      no visit | 0 | 81 | 0.0 |
|                        visit | 0 | 319 | 100.0 |
|          Overall Percentage | | | 79.8 |

a. Constant is included in the model. b. The cut value is 0.500.

**Table A3.** Variables in Equation.

| | B | S.E. | Wald | df | Sig. | Exp (B) |
|---|---|---|---|---|---|---|
| Step 0 Constant | 1.371 | 0.124 | 121.374 | 1 | 0.000 | 3.938 |

**Table A4.** Omnibus Tests of Model Coefficients.

| | | Chi-Square | df | Sig. |
|---|---|---|---|---|
| Step 1 | Step | 29.329 | 5 | 0.000 |
| | Block | 29.329 | 5 | 0.000 |
| | Model | 29.329 | 5 | 0.000 |

**Table A5.** Model Summary.

| Step | -2 Log Likelihood | Cox and Snell R Square | Nagelkerke R Square |
|---|---|---|---|
| 1 | 373.750 [a] | 0.071 | 0.111 |

[a]. Estimation terminated at iteration number 5 because parameter estimates changed by less than 0.001.

**Table A6.** Hosmer and Lemeshow Test.

| Step | Chi-Square | df | Sig. |
|---|---|---|---|
| 1 | 15.606 | 8 | 0.048 |

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
