# Peer review of "Influence of Spatial Accessibility and Environmental Quality on Youths’ Visit to Green Open Spaces (GOS) in Akure, Nigeria"

_sustainability, doi:10.3390/su151713223_

Round 1

Reviewer 1 Report (New Reviewer)

This is an interesting and important article about the "Influence of spatial accessibility and environmental quality on Youths' Visits to green open spaces in the city core, Akure Case, Nigeria." As such, this contributes to the literature and should be highlighted even more. However, there are some amendments that I suggest would strengthen the paper.

Comments

·         The findings of the study should be clearly stated in the abstract.

·         The novelty of this study could be articulated more in the abstract and conclusion.

·         The methodology is not clearly defined. How were the case study sites selected?

I would like to know more about the observations (were they made the whole year in all parks? How many days? Did each park get inspected once, or did you check several times during the study period to calculate the naturalness?

How were these survey participants recruited? What are the response rates?

How many research assistants were involved in conducting the survey on-site?

·         The conclusion section needs to be added, and some essential information that is supposed to be in the conclusion part needs to be included. For example, what are the findings to support the study's Aim/objectives?

·         Potential limitations and weaknesses of this study are missing.

·         Contribution to academia needs to be highlighted as well.

·         Relatedly, there is limited information about the reliability and validity of the study. This is important for the replicability of the process and the assurance of the findings.

·         English editing need improvement. See highlighted blue text in the pdf file.

Author Response

Reviewer 2 Report (New Reviewer)

I reviewed the whole manuscript. Too much writing errors, which make me cannot understand the paper. My comments are that the authors should improving the writing-English firstly. I just read the first 24 lines and found seven writing errors as following.

I cannot understand the paper because of the poor writing. My suggestion is that the paper should be rejected. At least, the author should improve the writing firstly to make the manuscript clarify.

Some commons are here.

1. Abstract. Lacking the conclusions of the results of this study.

2. Figure 2. This journal is facing to international readers. Figure 2 display the location of the study area in the country. However, this figure should exhibit the global location of the study area. Meanwhile, the left yellow polygon did not represent anything.

Some writing errors are as following.

1. Line 10: Urban development of recent has changed the lifestyle of city residents to sedentary. Of recent --> recently

2. Line 13: the planning and design --> the optimized planning and design of urban green space

3. Line 16. Findings --> The results

4. Line 20. Clarify the term "dynamics of urban living".

5. Line 21. Residents in cities now imbibe sedentary behaviour with a reduction in physical activities. Sedentary behaviour = reduction in physical activities. The logic of this sentence is confused.

6. Line 24. behaviour. (1,2) --> behaviour (1,2). one more period.

7. Line 22. activities.A --> activities. A. Lacking a space.

8. Line 174. aggegrate --> aggregate.

English very difficult to understand/incomprehensible

Author Response

Reviewer 3 Report (New Reviewer)

Dear authors,

I am glad to be a reviewer of such an interesting work about visit open green space by youth. However, I have a few suggestions for improving some parts of your manuscript:

1. please improve your Abstract. In this part of manuscript, the authors must clearly present the main goals of their work, so I encourage you to improve Abstract in that way.

-Figure 2-please provide a wider situation on map (my suggestion is to add a map of Africa as a continent)

Best regards

Round 2

Reviewer 2 Report (New Reviewer)

The core conclusion of the study is that the visits of youths to green open spaces (GOS) in Akure, Nigeria are influenced by spatial accessibility and environmental quality factors. The study found that factors such as age, street integration, social integration, and proximity of homes to GOS significantly influenced the visits of youths to GOS. The study also highlighted the negative impact of inadequate green open spaces on the physical and mental health of youths. Therefore, the study suggests that urban planning and policy should focus on improving the accessibility and environmental quality of green open spaces to meet the needs of youths and promote their health and well-being.

The core research method of this study was the use of a survey questionnaire to collect data. The researchers used a sample size of 399 participants to investigate the visits of youths to green open spaces in the core area of Akure, Nigeria. They analyzed the data using statistical analysis methods to examine the influence of factors such as age, street integration, social integration, and proximity to homes on youths' visits to green open spaces. The researchers also employed spatial analysis methods to assess the spatial accessibility and environmental quality of the green open spaces. Through these methods, the researchers identified the factors influencing youths' visits to green open spaces and drew conclusions based on their findings.

The main advantages of this article are that it investigates the influence of spatial accessibility and environmental quality on youths' visits to green open spaces in Akure, Nigeria. It also identifies significant factors such as age, street integration, social integration, and proximity to homes that influence visits to green open spaces. The study provides implications for urban planners and policymakers to design pedestrian-friendly networks and locate green open spaces in close proximity to residents' homes.

The main disadvantage of this article is that it does not capture all variables that may influence youths' visits to open spaces. It acknowledges the need for further research to explore other variables and situational circumstances that may impact visits to green open spaces.

Some minor comments are following.

Line 41 to 42. Following this, developing a viable urban green open space (GOS) has been proposed for. The last sentence of the first paragraph is not complete.

Equation 1. The constant e often stands for the natural constant, which is equal to 2.718281. The author should change another symbol to represent the confidence level and precision, but not use the latter e.

Extensive editing of the English language required

Round 3

Reviewer 2 Report (New Reviewer)

I suggest that this paper can be accepted after language checking. 

Minor editing of English language required

This manuscript is a resubmission of an earlier submission. The following is a list of the peer review reports and author responses from that submission.

Round 1

Reviewer 1 Report

This is an interesting manuscript about Influence of spatial accessibility and environmental quality on youths’ visit to green open spaces (GOS) in a country of western Africa. I read it with interest. The topic is very original with implication bot at social and environmental level. Perhaps, some more sentence about the ecological implication of this phenomenon could be added. Moreover, I think that the tpoic of social connectivity shouuld be matched with the analogous topic of ecological connectivity. Indeed people dynamism among sites may be assured when there is a co-occurrence of ecological dynamism among habitat patches in the landscape. However, I think that this ms deserves to be published on Sustainability after MINOR REVISIONS. I added some minor comments and suggestions that, I hope, could improve a bit the first draft of the manuscript. I would like re-read a further revised version of this origianl paper.

MINOR POINTS AND COMMENTS

Methods are strong and well-written.

Row 58. I would read some more sentences about the nature deficit disorder (see the paper of Louv and Driessnack, M. "Children and nature-deficit disorder." Journal for Specialists in Pediatric Nursing 14, no. 1 (2009): 73)

Rows 70 and 76. See for planning deisgn the concept of ecological network. This concept may be applied also to people inhabiting in suburban and rural contexts where habitat fragmentation make difficult the movements among towns and green areas (see, for Africa, the review on Afr. J. Ecol. https://doi.org/10.1111/aje.13186). In this paper the role of people in maintain ecological connectivity has been reported. This point could facilitate a match between social and ecological topics. See also the concept of spatial connectivity (row 92 and others).

Rows 106-108. Check for the correctedness of style in paragraph

Paragraph 3.1 is very interesting and well written.

About ‘naturalness’, I suggest to add the presence of animal biodiversity as variable.

In methods, I suggest to add some seminal handbook in statistics (for stat analyses): for example, Zar, Fowler and Cohen, Sokal and Rohlf, Dytham, etc.

Row 309. Add a reference for the SPSS software (which version has been used?).

Font in formula (row 335-336) should be smaller.

In Tab. 4, I suggest to add only 3 decimals.

Tab. 5. You should transform ‘4.735E+16’ with a number with exponent in apex.

Row 461. I suggets that the sentence ‘Some of the key findings of this research have been identified for further discussion’ is redundant and may be deleted.

Add the role of anonymous reviewers.

Have a nice work.